# Ketogenic Diet and Multiple Health Outcomes: An Umbrella Review of Meta-Analysis

**DOI:** 10.3390/nu15194161

**Published:** 2023-09-27

**Authors:** Shiyun Chen, Xin Su, Yonghui Feng, Ruojie Li, Minqi Liao, Laina Fan, Jiazi Liu, Shasha Chen, Shiwen Zhang, Jun Cai, Sui Zhu, Jianxiang Niu, Yanbin Ye, Kenneth Lo, Fangfang Zeng

**Affiliations:** 1Department of Public Health and Preventive Medicine, School of Medicine, Jinan University, No. 601 Huangpu Road West, Guangzhou 510632, China; chenshiyun@stu2022.jnu.edu.cn (S.C.); suxin@stu2021.jnu.edu.cn (X.S.); fyh1001@stu2022.jnu.edu.cn (Y.F.); lrj1999@stu2022.jnu.edu.cn (R.L.); 202235821030liu@stu2022.jnu.edu.cn (J.L.); chenssjnu@126.com (S.C.); zswvera@stu2021.jnu.edu.cn (S.Z.); caijunjnu2021@stu2021.jnu.edu.cn (J.C.); zhusui1213@jnu.edu.cn (S.Z.); 2Institute of Epidemiology, Helmholtz Zentrum Munich-German Research Center for Environmental Health, Ingolstadt Landstr. 1, 85764 Neuherberg, Germany; minqi.liao@helmholtz-muenchen.de; 3Department of Clinical Medicine, International School, Jinan University, No. 601 Huangpu Road West, Guangzhou 510632, China; lainafan@stu2020.jnu.edu.cn; 4General Surgery, The Affiliated Hospital of Inner Mongolia Medical University, No. 1 Tongdao North Road, Hohhot 010000, China; nu3-2000@163.com; 5Department of Clinical Nutrition, The First Affiliated Hospital, Jinan University, Guangzhou 510630, China; yeyanbin2023@jnu.edu.cn; 6Department of Food Science and Nutrition, The Hong Kong Polytechnic University, Kowloon, Hong Kong 100872, China; 7Research Institute for Future Food, The Hong Kong Polytechnic University, Kowloon, Hong Kong 100872, China

**Keywords:** ketogenic diets, obesity, overweight, seizure reduction, cancer, umbrella review, meta-analysis

## Abstract

Numerous studies have examined the effects of ketogenic diets (KD) on health-related outcomes through meta-analyses. However, the presence of biases may compromise the reliability of conclusions. Therefore, we conducted an umbrella review to collate and appraise the strength of evidence on the efficacy of KD interventions. We conducted a comprehensive search on PubMed, EMBASE, and the Cochrane Database until April 2023 to identify meta-analyses that investigated the treatment effects of KD for multiple health conditions, which yielded 23 meta-analyses for quantitative analyses. The evidence suggests that KD could increase the levels of low-density lipoprotein cholesterol (LDL-C), total cholesterol (TC) and high-density lipoprotein cholesterol (HDL-C), the respiratory exchange rate (RER), and could decrease total testosterone and testosterone levels (all *p*-random effects: <0.05). The combination of KD and physical activity can significantly reduce body weight and increase the levels of LDL-C and cortisol. In addition, KD was associated with seizure reduction in children, which can be explained by the ketosis state as induced by the diet. Furthermore, KD demonstrated a better alleviation effect in refractory childhood epilepsy, in terms of median effective rates for seizure reduction of ≥50%, ≥90%, and seizure freedom. However, the strength of evidence supporting the aforementioned associations was generally weak, thereby challenging their credibility. Consequently, future studies should prioritize stringent research protocols to ascertain whether KD interventions with longer intervention periods hold promise as a viable treatment option for various diseases.

## 1. Introduction

The ketogenic diet (KD) was defined as a diet that induces ketosis, which is characterized by a very low carbohydrate content, limited to 5–10% of total daily calorie intake [1,2]. KD was first proposed in the 1920s for the treatment of diabetes and epilepsy [3], particularly refractory childhood epilepsy [1,2], and it emphasizes a ratio of fat to carbohydrate and protein, typically in the ratio of 4:1 or 3:1 [3]. 

In the present day, there exist four primary types of ketogenic dietary therapies (KDTs): the classic ketogenic diet, the modified Atkins diet (MAD), the medium-chain triglyceride KD (MCTKD), and the low glycemic index treatment (LGIT) [4]. Additionally, KD, encompassing the classical ketogenic diet and the modified Atkins diet, is considered two commonly employed dietary approaches for treating intractable epilepsy worldwide [5].

Studies have shown that, under a low-carbohydrate and high-fat diet, glycerol produced from triglyceride (TG) hydrolysis could generate 16–60% glucose in the liver to directly supply a small amount of energy to the central nervous system in the early stage [6], which was beneficial in reducing blood glucose and insulin concentration, and increasing insulin sensitivity [7,8]. In addition, KD can increase the level of ketone bodies (KB), as an intermediate product produced by fat oxidizing and decomposing in the liver, [3], which can directly suppress appetite and reduce calorie intake [9]. Furthermore, KD can also inhibit the production of inflammatory factors (including NLRP3 and NF-κB) and activate GRP109A to reduce inflammatory response [3]. Thus, KD has been promoted as a weight-loss diet [10,11] and as a treatment for obesity-related diseases, such as hypertension [12,13], cardiovascular disease (CVD) [14], cancer [15,16], and sleep apnea [17]. In addition to dietary interventions, exercise is another important weight-loss strategy by increasing energy expenditure and decreasing body fat accumulation. Multiple studies have examined the impact of combining physical activity with KD on body weight, blood lipids, cortisol, cardiorespiratory function, and epilepsy [18,19]. However, the effects of KD only or KD combined with physical activity on the different outcomes are still controversial. 

Another concern about the use of KD is its impact on blood lipids. Animal studies have shown that KD can lead to lipid abnormalities [20,21], but findings from population-based epidemiological studies have been inconsistent [22,23,24,25]. It is well-known that lipid abnormalities increased the risk of cardiovascular diseases, such as atherosclerosis [26]. When consuming KD, a large amount of fatty acids is metabolized in the liver to produce ketones, such as β-hydroxybutyrate, which enter the bloodstream and lead to a condition called ketosis, with serum beta-hydroxybutyrate >3 mmol/L [27]. One study of women with ovarian and endometrial cancer who have received KD intervention showed a significantly higher concentration of blood β-hydroxybutyrate in their blood [28]. Another case report indicated that a woman without a history of diabetes experienced ketoacidosis four times after using a low-carbohydrate diet, but she no longer developed ketoacidosis after resuming normal carbohydrate intake [29]. When following a ketogenic diet, human body does not have enough glucose in the blood or liver, leading to the production of special compounds called ketones; healthy people have enough insulin in their bodies to suppress ketone production, but in the case of diabetes (including type 1 diabetes and T2DM) [30,31], alcoholism, hunger or extreme dieting [32], and some metabolic diseases [33], ketosis can lead to excess ketones in the body, which will lead to ketoacidosis. Therefore, we should consider its adverse effects when KD is used in clinical practice.

To date, numerous meta-analyses have been conducted to assess the effects of KD intervention on health-related outcomes. However, there are various biases (e.g., number of participants, publication bias, reporting bias, residual confounding bias, and researcher allegiance). Therefore, this umbrella review aims to systematically collect meta-analyses on the efficacy of KD in managing diverse health problems, and to evaluate the reliability and strength of evidence for these interventions.

## 2. Methods

Protocol and registration details for this umbrella review have been pre-registered with PROSPERO (International Prospective Register of Systematic Reviews) under the registration number CRD42023413116. The study followed the Preferred Reporting Items for Systematic Reviews and Meta-analyses (PRISMA) reporting guideline [34].

### 2.1. Search Strategy

Electronic databases, including PubMed, Cochrane Database, and EMBASE of Systematic Reviews, were searched from inception until 2 April 2023, for eligible meta-analyses of RCTs of KD interventions on various outcomes. The Appendix A have provided detailed information on search strategies. We also manually searched the reference lists from relevant studies to reduce missing records in database searches. If there were any discrepancies, a third investigator (FZ) was consulted, and a consensus was reached through discussion.

### 2.2. Inclusion and Exclusion Criteria

The identified studies were independently reviewed by two authors based on the following inclusion criteria: (1) a published meta-analysis or systematic review and meta-analysis in the English language; (2) interventions should involve a KD alone or in combination with exercise, and KD was defined as having 5–10% of total energy from carbohydrates, without specifying the exact amounts of fat and protein (with fat being higher than protein) or should have a carbohydrate content below 50 g/day [35]. Additionally, various forms of KD have been considered, including the classic ketogenic diet, the modified Atkins diet (MAD), medium-chain triglyceride KD (MCTKD), and low glycemic index treatment (LGIT); (3) health outcomes should include changes in indicators such as blood lipid markers and weight, or improvements in disease risk such as seizure reduction; and (4) the summary effect size with 95% confidence intervals (CI) must be reported. Studies that did not conduct a meta-analysis or a systematic review and animal studies were excluded. Additionally, studies that did not clearly describe intervention measures were also excluded. All discrepancies between the two authors were discussed and resolved by consensus. Furthermore, if the same population (from the same original study) was used in several articles, only the most recent, complete, or largest study was included in the umbrella review. However, if different outcomes were reported for this population in different meta-analyses, all outcomes would be included in the umbrella review [36].

### 2.3. Data Extraction and Estimate of Methodological Quality

Two investigators (CSY and LYJ) independently extracted data from the studies, including the first author’s name, year of publication, number of studies included, sample size, type of population, type of outcomes, definition of the KD, and summary meta-analytic estimates. For each study, the following information was extracted: year of publication, study design (i.e., clinical trial, cohort, or case-control), population type (i.e., individuals with type 2 diabetes, obesity, or pediatric seizures), sample size, type of KD reported, study outcomes, and maximally adjusted study-specific estimates, which were mean difference (MD), standardized mean difference (SMD), odds ratio (OR), or relative risk (RR), with 95% confidence intervals (CIs).

We assessed the methodological quality of each meta-analysis using the Assessment of Multiple Systematic Reviews, version 2 (AMSTAR-2) tool, a 16-item rating scale with good interrater reliability and usability. The AMSTAR 2 quality assessment categorizes meta-analyses into four levels: critically low, low, moderate, or high. Of these 16 areas, 7 can particularly influence the validity of the review and its conclusions and are considered “key areas” (2) a priori design provided; (4) comprehensive literature search; (7) presence of a list of excluded studies, along with reason for exclusion; (9) risk of bias assessment; (11) methods for statistical combination of results; (13) discussion/interpretation of the potential impact of risk of bias of individual studies on the meta-analysis result; (15) likelihood of publication bias [37]. 

A high-quality meta-analysis should not have any critical defects and at most one non-critical weakness. On the other hand, a moderate quality meta-analysis may have more than one non-critical weakness but should still lack any critical defects. If a meta-analysis has a critical flaw, it is considered low-quality, regardless of its non-critical weaknesses. Finally, if a meta-analysis exhibits more than one critical defect, it falls under the critically low-quality category, and these defects might also exist alongside non-critical weaknesses.

### 2.4. Statistical Analysis

We followed the methodology developed in a previous umbrella review to conduct our analysis [38,39,40,41,42]. Pooled effect sizes, 95% confidence intervals (CI), and *p*-values for each meta-analysis were re-estimated in raw form using the DerSimonian and Laird method under a random-effects model [43]. The *p*-value cutoff for pooled effect estimates was set at <0.05, and additional *p*-value cutoffs were set at <10^−3^ and <10^−6^ to assess the reliability of the evidence [44,45]. To detect heterogeneity between studies, we performed the Cochrane Q test (with *p* < 0.10 considered statistically significant) [46], and calculated the *I*^2^ statistic with *I*^2^ ≥ 50% indicating high inconsistency [47]. 

We also calculated 95% prediction intervals, which indicate the uncertainty of the effect that would be anticipated in a subsequent study examining the same association, while taking into account study heterogeneity [48,49]. Prediction intervals that do not include the null value (i.e., 1 in the case of ORs or RRs) suggest that the effect would be expected in a subsequent study [48,49]. We assessed publication bias by using an Egger test [50]. Egger’s *p*-value less than 0.10 demonstrated the presence of small-study effects, with the estimate of the largest component study (i.e., the study with the smallest SE) being more conservative than the random-effects model summary estimate [38,39,40,41,42].

We evaluated the excess significance to examine whether the observed number of studies (O) with nominally statistically significant results (*p* < 0.05) in each meta-analysis was larger than the expected number (E) [51]. For each meta-analysis, we estimated the expected number of significant studies by summing the statistical power estimates for each individual study [42], using an algorithm from a noncentral t distribution and the effect size of the largest study in each meta-analysis as the plausible power for the tested association [52]. The significance threshold for excess significance bias was set at *p* < 0.10 for each meta-analysis. We used *p* < 0.10 (one-sided *p* < 0.05 with O > E, as previously proposed) to determine excess significance for a single meta-analysis [51]. 

### 2.5. Assessment of Evidence Credibility

The following criteria were used to determine the level of evidence [53,54,55]: (1) *p* < 10^−6^ based on random-effects meta-analysis; (2) >1000 participants; (3) *p* < 0.05 of the largest/study; (4) between-study heterogeneity with *I*^2^ < 50%; (5) no evidence of small-study effects; (6) 95% prediction interval that excluded the null value; and (7) no excess significance bias. 

Based on the criteria outlined above, we classified the level of evidence for each association into one of five categories: convincing (Class I), highly suggestive (Class II), suggestive (Class III), weak (Class IV), and not significant (NS). If all seven criteria were met, the evidence was classified as convincing. If 1–3 criteria were met, the evidence was classified as highly suggestive. If only the criterion of *p* ≤ 0.001 by random-effects and >1000 participants was met, the evidence was classified as suggestive. If only the criterion of *p* ≤ 0.05 under random-effects was met, the evidence was classified as weak. If the *p*-value was greater than 0.05 under random-effects, the evidence was classified as not significant.

Statistical analyses were performed using Stata, version 15.0 (Stata Corp, College Station, TX, USA). Apart from the cutoff value mentioned above, the significance level was set at 0.05 (2-tailed).

## 3. Results

A total of 1942 records were identified. After removing duplicates and screening titles and abstracts, 810 articles were excluded, and 108 references were considered for full-text evaluation. Finally, 23 studies comprising 149 comparisons were included in this umbrella review (Figure 1) [5,9,18,24,56,57,58,59,60,61,62,63,64,65,66,67,68,69,70,71,72,73,74]. Excluded studies with reasons for exclusion are presented in Appendix A. Table 1 and Table 2 show the details of the included systematic reviews.

The quality of the reviews was assessed using AMSTAR-2, 3 with high-quality ratings [64,65,67], 10 with low-quality ratings [5,18,24,56,58,59,64,66,73,74], and 10 with critically low-quality ratings (Table 1 and Table 2) [9,57,60,61,62,67,69,70,71,72].

A total of 117 comparisons were reviewed to assess the efficacy of KD or MAD in health indicators, among participants with diseases such as obesity, diabetes, seizures, cancer, or healthy individuals. Of these, only 12 [24,60,63,64,65,67,68,74] comparisons showed statistically significant results based on the random-effects model, indicating limited overall evidence for the effectiveness of KD or MAD. Three of these significant results had 95% prediction intervals that excluded the null value. Comparisons with statistically significant results generally had no or low heterogeneity, except for the RER outcome indicators after KD intervention in endurance athletes, which showed high heterogeneity. A small-study effects bias was observed in 51 comparisons, and excess significance bias was detected in 16 comparisons. Notably, 61 comparisons had fewer than five included studies, which may have reduced the power of the analysis.

Based on the quantitative criteria used in this umbrella review, none of the meta-analyses provided convincing or highly suggestive evidence of association. Only weak evidence was observed in the 10 meta-analyses reporting a significant association (*p* < 0.05) (Figure 2 and Figure 3).

A total of 32 comparisons assessed the effects of combined KD and physical activity on related indicators in people with overweight and obesity, healthy adult males, and athletes. Among them, only seven [61,63] comparisons reported a marginally statistically significant result, with five 95% prediction intervals excluding the null value (Appendix A). Moreover, 12 comparisons showed a risk of small-study effects bias, while 4 comparisons were detected with excess significance bias.

There were two main types of dietary interventions, KD and MAD, while physical activity was classified based on the type of exercise, such as resistance training, progressive resistance training, high-intensity interval training, medium-intensity continuous training, and hypertrophy training. Physical activity was also classified based on duration, such as long-term exercise (≥20 min) and short-term exercise (<20 min).

### 3.1. KD or KD Combined with Physical Activity among Healthy Individuals

A study of 103 endurance athletes on a dietary intervention showed significant overall differences in RER between the K-LCHF and HC/HD diets (SMD: −1.81, CI: −2.49, –1.13, *p* < 0.00001), with a high degree of heterogeneity (*I*^2^ = 58%) [63] (Figure 2 and Appendix A). Despite the high heterogeneity, the included studies showed a significant decrease in RER after the intervention. 

In addition to KD, two variations of KD, namely high-protein KD (HP-KD) and moderate-protein KD (MP-KD), have been identified based on their protein content. Scientific evidence suggests that HP-KD can decrease total testosterone levels in healthy adult men at rest (SMD: −1.04; *p*-random effects: 0.003). Intriguingly, even a short-term implementation of KD lasting less than 3 weeks can reduce total testosterone levels (SMD: −1.04; *p*-random effects: 0.003) [63] (Figure 2 and Appendix A). In addition, long-term exercise in healthy adult males leads to a significant increase in cortisol levels at both 0 *h* (SMD: 0.705; *p*-random effects: <0.001) and 1 h (SMD: 0.65; *p*-random effects: 0.004) post-exercise. Cortisol levels also increased under several conditions, including sustained implementation of KD for less than three weeks and immediate measurement after exercise (SMD: 0.5; *p*-random effects: 0.018), implementation of MP-KD and immediate measurement after exercise (SMD: 0.488; *p*-random effects: 0.047), and prolonged implementation of MP-KD and measurement at two hours post-exercise (SMD: 0.818; *p*-random effects: 0.001) [63] (Figure 4 and Appendix A). 

### 3.2. KD or KD Combined with Physical Activity and Obesity or Overweight Individuals with or without Diabetes

Notably, weak evidence suggested that KD intervention may lead to a significant increase in low-density lipoprotein (LDL) (SMD: 0.349; *p*-random effects: 0.009) and total cholesterol (TC) (SMD: 0.381; *p*-random effects: 0.012) levels in individuals with overweight or obesity but without T2DM [24] (Figure 2 and Appendix A). For individuals with overweight or obese, a combination of KD and physical activity can significantly reduce body weight (SMD: −0.301; *p*-random effects: 0.016) and increase LDL levels (SMD: 0.736; *p*-random effects: 0.002) [61] (Figure 4 and Appendix A).

Previous studies have shown that taking an LCKD has beneficial effects on blood sugar control, weight and the use of hypoglycemic drugs in patients with type 2 diabetes [75]. For a low-carbohydrate, high-fat diet, is generally associated with higher concentrations of LDL and HDL cholesterol and lower serum concentrations of triglycerides than is the conventional intake of carbohydrates and fat [76]. In our study, it was shown that intervention with KD in people with type 2 diabetes leads to an increase in HDL cholesterol (MD: 0.283; *p*-random effects: 0.003) and a decrease in TG levels (MD: −0.367; *p*-random effects: <0.001) [64] (Figure 2 and Appendix A).

### 3.3. KD and Seizures or Epilepsy Reduction

Ten meta-analyses [5,67,68,69,70,71,72,73,74] were conducted to evaluate the impact of KD or MAD on the proportion of seizure reduction observed in various conditions such as pediatric epilepsy, childhood epilepsy, refractory epilepsy, Lennox–Gastaut syndrome, Dravet syndrome, infantile spasms, infant epilepsy, CDKL5-related epilepsy, drug-resistant epilepsy, and Sturge–Weber syndrome (Figure 3, Appendix A). 

KD and/or MAD interventions were found to significantly reduce the relative risk (RR) of ≥50% seizure reduction in children (pooled RR: 4.776; *p*-random effects: <0.001) [67], with further evidence indicating that KD use reduced the odds ratio (OR) of ≥50% seizure reduction in children (pooled OR: 4.641; *p*-random effects: <0.001) [73] (Figure 3 and Appendix A). Lastly, weak evidence suggested that KD use may reduce the odds ratio of definite responder rate in CDKL5-related epilepsy (pooled OR: 0.225; *p*-random effects: 0.012) [74] and may reduce the RR of ≥50% seizure reduction: children (pooled RR: 5.641; *p*-random effects: <0.001) in Drug-resistant epilepsy [68] (Appendix A). For patients with Dravet syndrome [5,69], the median effective rate for ≥50% seizure reduction (≥50%) in the KD study at the 3rd, 6th, and 12th months were 62.8%, 60.2%, and 61.3% from baseline (*n* = 191). 

Similarly, for infants with epilepsy on KD [71], the median effective rates for ≥50% and seizure-free (SF) outcomes in the third month were 69.0% and 36.7%, respectively (*n* = 430). Furthermore, two studies [5] (*n* = 66) on infantile spasms reported promising results with the median effective rates for ≥50%, ≥90% reduction in seizures (≥90%), and SF in the 3rd month being 65.3%, 11.3%, and 19.6%, respectively. In the 6th month, the rates were reduced to 55.0%, 9.9%, and 21.9%, while in the 12th month, the rates were 56.5%, 8.5%, and 4.2%, respectively. Two studies comprising a total of 118 participants have reported encouraging findings regarding the treatment of Lennox–Gastaut syndrome [5]. Specifically, the median effective rates for ≥50% in the 3rd month was 61.8%; the rates for ≥50%, ≥90%, and SF in the 6th month were 47.7%, 20.8%, and 5%, respectively.

A total of 59 original articles were included that reported on refractory childhood epilepsy [5], comprising 47 articles on the KD (*n* = 2874) and 12 articles on the MAD (*n* = 404). For studies that employed the KD as an intervention, the median effective rates for seizure reduction of ≥50%, ≥90%, and seizure freedom at the 3rd 6th, and 12th months were reported as follows: month-3: 60.6%, 40.0%, and 26.6%; month-6: 52.6%, 33.5%, 20.1%; month-12: 42.8%, 27.6%, 14.0%. Similarly, for studies that implemented MAD, the median effective rates for seizure reduction of ≥50%, ≥90%, and seizure freedom at month-3 were 47.7%, 25.5%, and 19.7%; at month-6, the rates for seizure reduction of ≥50% and seizure freedom were 43.3% and 20.8%, respectively; at month-12, the rates for seizure reduction of ≥50% and seizure freedom were 25.1% and 9.2%, respectively. Furthermore, six articles (*n* = 174) that used the MAD reported the median effective rates for seizure reduction of ≥50%, ≥90%, and seizure freedom in refractory adult epilepsy [72] at month-3, which were 40.5%, 13.6%, and 9.9%, respectively. The rate for seizure reduction of ≥50% at month 6 and month 12 were 36.4% and 30.3%, respectively, while the rates for seizure freedom were 10.0% and 13.3%, respectively.

### 3.4. KD and Cancer

Two meta-analyses [60,66] (*n* = 652) were conducted to evaluate the impact of KD on cancer patients. In cancer patients, there is no significant difference in fasting glucose (SMD: −0.402; *p*-random effects: 0.342), fasting insulin (SMD: 0.107; *p*-random effects: 0.884), HDL (SMD: −0.185; *p*-random effects: 0.646), LDL (SMD: 0.211; *p*-random effects: 0.321), TC (SMD: 0.248; *p*-random effects: 0.245), TG (SMD: 0.152; *p*-random effects: 0.638), RR of adverse events (pooled RR: 1.263; *p*-random effects: 0.75) [67], body mass index (BMI) (WMD: −1.401; *p*-random effects: 0.797), body weight (WMD: −0.215; *p*-random effects: 0.922) (Figure 2 and Appendix A) [65]. Moreover, KD interventions were found to increase the incidence of ketosis (pooled RR: 3.578; *p*-random effects: 0.01) [60] (Figure 2 and Appendix A).

## 4. Discussion

Our study provides an umbrella review of meta-analyses evaluating the effects of the KD on health outcomes among various populations, including healthy individuals, as well as people with overweight or obesity, and patients with chronic conditions such as type 2 diabetes, cancer, and epilepsy. We examined a range of outcomes such as weight loss, appetite, epilepsy, adverse events, and blood indicators including blood glucose and lipids. However, our findings must be interpreted with caution due to the weak strength and credibility of the associations and evidence presented. One of the main reasons for this is the small sample size of the studies included.

### 4.1. The Effect of KD or KD Combined with Physical Activity among the General Population

Weak evidence showed that KD significantly reduced RER. RER refers to the ratio of carbon dioxide exhaled by the human body to oxygen inhaled, indirectly showing the muscle’s oxidative capacity to get energy [77]. Compared with the energy supply of glycogen, the metabolism of ketone bodies requires less oxygen, thus reducing RER.

The combination of KD and exercise has been found to maintain high cortisol levels in men within 2 h after exercise, but KD alone does not appear to affect cortisol levels [64], which may be a normal physiological response to exercise [78]. However, intervention with KD alone can lead to a decrease in testosterone levels in men at rest, and studies have shown that men with low total testosterone levels have a 44% higher risk of mortality, as well as an increased risk of specific cardiovascular and respiratory diseases by more than 38% [79]. Therefore, we need to be vigilant about the potential endocrine hormone level changes caused by low-carbohydrate diets. It is important to note that the number of articles included in this study is small, so more studies with larger populations are needed to verify these findings.

### 4.2. The Effect of KD or KD Combined with Physical Activity among the Population with Metabolic Risk Factors

Weak evidence shows that KD intervention in patients with T2DM can significantly increase HDL and decrease TG [64], while there is weak evidence KD alone/KD combined with exercise intervention on people with overweight or obese shows an increment in LDL [24,61] and TC [24]. High levels of HDL are associated with a reduced risk of cardiovascular disease [80], while high levels of TC/LDL/TG are major risk factors for cardiovascular disease [81]. However, the evidence we have found suggests that there is a paradoxical relationship between reducing risk and increasing risk, and a possible explanation for this phenomenon is the type of fat intake. Studies have shown that increasing the intake of high saturated fatty acids can lead to an increase in low LDL levels while increasing intake of high unsaturated fatty acids can lead to a decrease in LDL levels [82]. In addition, polyunsaturated fatty acids can also lower TC and TG levels and increase HDL levels [82]. A previous randomized controlled trial found that a low-carbohydrate diet rich in unsaturated fatty acids but low in saturated fatty acids greatly influenced lipid distribution and substantially improved glycemic control and CVD risk biomarkers [83]. The other possible reasons for the decrease in TG levels were as follows. First, a high-fat diet can promote the body’s metabolism from carbohydrate burning to fat burning, promote fat oxidation and metabolism, and reduce the rate of synthesis of TG, thereby lowering TG levels [84]. Second, improved insulin sensitivity leads to reduced insulin release [3]. Insulin is a hormone that regulates fat metabolism and promotes fat synthesis and storage in the body [85]. By lowering insulin levels, KD can reduce the synthesis and accumulation of TG. In addition, it should be noted that although LDL is widely regarded as a risk factor for cardiovascular disease, in clinical practice, plasma LDL levels are usually not directly measured, but estimated based on its cholesterol concentration (LDL-C), and in most cases, LDL-C concentration and LDL particle number are highly correlated. However, in certain situations (such as metabolic syndrome, and diabetes), small, dense, low-cholesterol LDL particles may predominate, and the measurement of plasma LDL-C and LDL particle concentration may become inaccurate [86]. Therefore, this requires us to focus not only on the concentration of LDL but also on the quality of LDL when considering its impact on cardiovascular disease.

In addition, we also observed weak evidence for the association between KD combined with exercise and weight loss in overweight and obese populations [61]. In the study conducted by Lee et al. [61], the intervention group received exercise combined with KD, and the control group received exercise combined with a conventional diet. In contrast, the intervention group and control group of other articles evaluating the effect of KD for weight loss were KD and conventional diet respectively [24,56,57]. Therefore, we can infer that exercise and KD may have a synergistic effect on weight loss. The main mechanism of KD for weight loss is to suppress appetite and reduce total energy intake by restricting carbohydrate intake and increasing ketone body levels [9]. This maximizes the oxidation of lipids from endogenous sources.

### 4.3. KD and Seizures or Epilepsy Reduction

Weak evidence showed that KDs were effective in the treatment of pediatric seizures [63], adult intractable epilepsy [72], drug-resistant epilepsy [68], and CDKL5-related epilepsy [74]. We also qualitatively summarized the effectiveness of KDs for treating different types of epilepsy, including Dravet syndrome, epilepsy in adults and adolescents, infant epilepsy, pediatric epilepsy, Lennox–Gastaut syndrome, and refractory adult/adolescent/childhood epilepsy. Although it is a qualitative description, our results showed that the efficacy of ketogenic diets in achieving ≥50% seizure reduction ranged from 8.9% (refractory childhood epilepsy) to 85% (Dravet syndrome). There is evidence to suggest that KDs demonstrated a higher effectiveness in treating generalized seizures compared to partial seizures [87]. Epilepsy is associated with abnormal nerve cell excitation, and improving the depolarization state of nerve cells may be a potential mechanism for KDs in treating epilepsy [3]. Studies have shown that injection of ketone bodies (KBs) reduced the susceptibility to seizures in mice, and histological sections of mice showed a decreased rate of spontaneous firing [88,89]. Additionally, KBs can directly inhibit vesicular glutamate transport [90]. The inhibitory neurotransmitter γ-aminobutyric acid (GABA), which is converted from glutamate, can reduce membrane excitability. Several human and animal studies have shown that KDs increase GABA and decrease glutamate in the brain [91,92]. Glycolysis may also play a role in promoting seizures. The metabolic capacity of glucose in the brain center increases the necessary energy for nerve cell excitation, while the production of KBs in KDs is a process of anaerobic metabolism that slows down energy supply, thereby reducing the onset of epilepsy [93]. An animal experiment used the glucose analog 2-deoxyglucose to reduce glucose uptake by brain nerves in mice and compete with phosphoglucose isomerase to inhibit glycolysis and slow down the progression of epileptic seizures [94]. It should be noted that, currently, KD is often recommended as an adjunctive or alternative treatment for epilepsy, due to the significant side effects of medication [87].

### 4.4. KD and Cancer

The incidence of KD induced ketosis in cancer patients is 3.6 times that of people who eat a general diet, which may imply that cancer patients are more responsive to KD. Cancer cells require a lot of energy to proliferate, and cancer cells take up more glucose than normal cells and convert glucose to lactate for energy through glycolysis, a process known as the Warburg effect [95,96]. As KD reduces glucose intake, it may make cancer cells unable to produce energy through glycolysis, and may help to treat cancer. In addition, the reduced glucose intake allows for the production of KBs, which will be utilized by normal cells. This leads to a decrease in insulin and insulin-like growth factors, which are known to be important for cancer cell proliferation [97]. In summary, the use of KD puts cancer cells in an unfavorable environment, which may enhance the antitumor effect. 

However, KD may not be effective in treating all cancers. In one study, a ketogenic diet was shown to affect BRAF V600E mutation-dependent MEK1 activation in a xenograft mouse model, contributing to increased human melanoma growth in this model. In addition, studies have reported side effects, including nausea, constipation, vomiting, acidosis, hypoglycemia, and fatigue in the short term [98,99,100,101], and long-term side effects such as hyperlipidemia, hypercholesterolemia, and bone mineral loss [102].

In conclusion, KD may be a potential treatment for certain types of cancer, but its effectiveness may vary across different cancer types. As of now, the lack of rigorous clinical trials examining the safety and efficacy of KD in cancer treatment calls for further research. It is important to conduct comprehensive clinical trials involving patients with various types of cancer to better evaluate the role of KD in treating cancer. There are still insufficient data to suggest that KD can be used as a primary anti-cancer therapy, but it may have a beneficial role as an adjuvant therapy.

#### Strengths and Limitations

Our research has several strengths. Firstly, we evaluated the efficacy of KD on multiple health-related outcomes. We have summarized outcome indicators that significantly improve or worsen after KD intervention, which can increase understanding of the impact of KD on multiple health-related outcomes. In addition, most of the studies we included used a randomized controlled trial design, which is usually considered to be of high quality. This can increase the credibility of the article and provide more reliable conclusions. However, our study had some limitations. First, we used the AMSTAR2 tool to assess the quality of the systematic reviews included in this study and found that the primary shortcomings were the lack of an excluded studies list, with insufficient investigation and discussion on publication bias. These may influence the credibility of our conclusion. As an umbrella review, we were unable to address any potential biases, which may have resulted in concluding with a higher risk of type I errors based on the reanalyzed data. Additionally, it should be noted that quality evaluation can be subjective. Although the vast majority of studies included in our article were randomized controlled trials, which are considered a high-quality research design, the sample size of included intervention studies was often small, with only 13,014 people. This became a key factor that led to a generally low level of evidence. Thirdly, inconsistencies in the type of KD (including traditional KD, MDA, and VLCKD with inconsistent nutrient ratios), population characteristics, and the design of original studies appear to be the sources of heterogeneity in intervention outcomes. Finally, the majority of articles included in our review assessed the short-term health outcomes of the KD intervention, but limited long-term follow-up data were available, which could reflect the effect of KD on cardiovascular, renal, gastrointestinal disease, and so on. As a result, there is a lack of evidence regarding the long-term efficacy of the KD, such as its ability to sustain weight loss over time or the potential for increased adverse health outcomes. 

## 5. Conclusions

Based on current findings, there is weak evidence that KD can be used as an effective treatment for epilepsy and effectively improve blood lipids among patients with T2DM. Additionally, the strength of evidence that KD combined with exercise reduced body weight in overweight and obese individuals and reduced respiratory exchange rates in athletes is weak. However, most interventions only examine short-term effects, and there is a need for more comprehensive investigations that examine the long-term implications, adherence, and potential side effects associated with KD. Furthermore, individual variations in responses to KD should be taken into account when considering its effectiveness in different populations.

## Figures and Tables

**Figure 1 nutrients-15-04161-f001:**
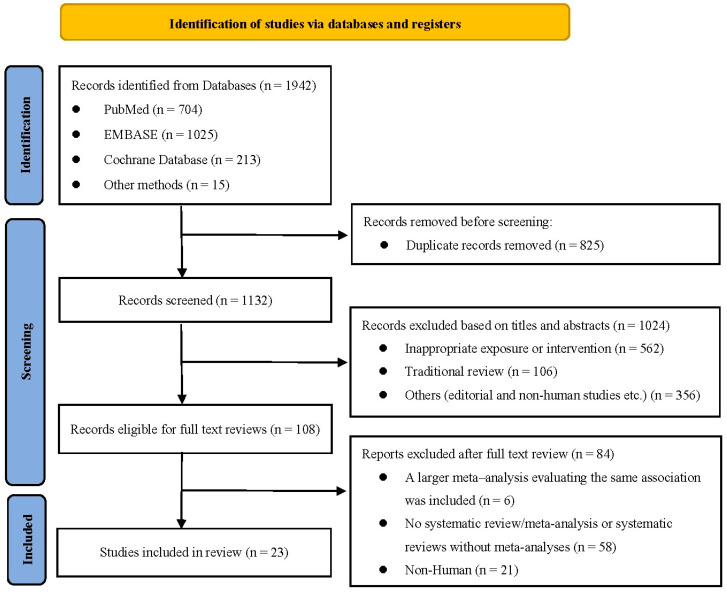
Study selection profile.

**Figure 2 nutrients-15-04161-f002:**
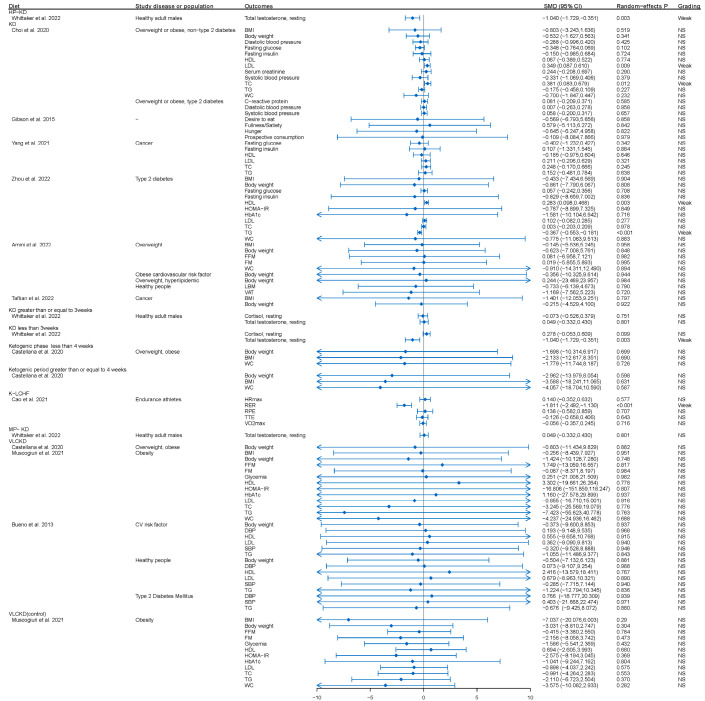
Quantitative synthesis and evidence grading for meta-analyses of diet intervention for clinical and non-clinical health problems. SMD, standardized mean difference [5,9,24,57,58,59,64,65,66,69,70,71,72]. BMI, body mass index; DBP, diastolic blood pressure; FFM, fat-free mass; FM, fat mass; HDL, high density lipoprotein; HOMA-IR, homeostatic model assessment index of insulin resistance; HbA1c, glycated hemoglobin; HRmax, maximal heart rate during exercise; LBM, lean body mass; LDL, low density lipoprotein; RER, respiratory exchange rate; RPE, perceived exertion; SBP, systolic blood pressure; TTE, time to exhaustion; TC, total cholesterol; TG, triglycerides; VAT, visceral adipose tissue; VO2max, maximum oxygen uptake; WC, waist circumference; KD, ketogenic diet; VLCKD, very-low-carbohydrate ketogenic diets; K-LCHF, ketogenic low-carbohydrate, high-fat; MAD, modified Atkins diet; HP, high-protein; MP, moderate-protein; MD, mean difference; O, observed number of studies with positive finding; ref, reference; SMD, standardized mean difference; WMD, weighted mean difference; OR, odds ratio; RR, relative risk.

**Figure 3 nutrients-15-04161-f003:**
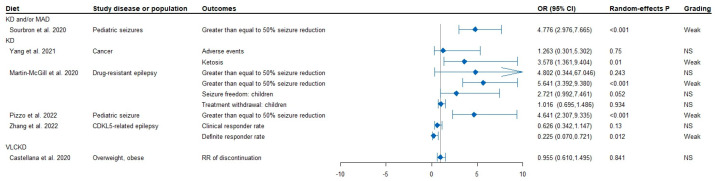
Quantitative synthesis and evidence grading for meta-analyses of diet intervention for clinical and non-clinical health problems OR, odds ratio [56,60,67,68,73,74]. KD, ketogenic diet; VLCKD, very-low-carbohydrate ketogenic diets; MAD, modified Atkins diet; OR, odds ratio.

**Figure 4 nutrients-15-04161-f004:**
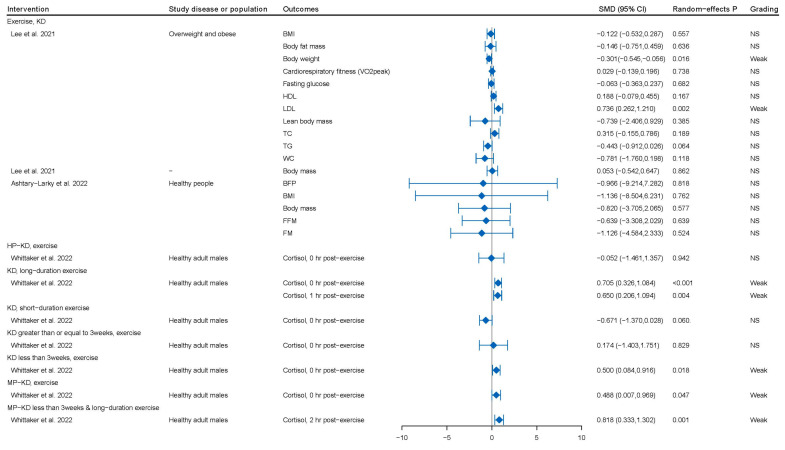
Quantitative synthesis and evidence grading for meta-analyses of the combination of diet and physical activity interventions for clinical and non-clinical health problems. SMD, standardized mean difference [18,61,62,63]. BFP, body fat percentage; BMI, body mass index; FFM, fat-free mass; FM, fat mass; HDL, high density lipoprotein; LDL, low density lipoprotein; TC, total cholesterol; TG, triglycerides; WC, waist circumference; KD, ketogenic diet; HP, high-protein; MP, moderate-protein; MD, mean difference; O, observed number of studies with positive finding; ref, reference; SMD, standardized mean difference; WMD, weighted mean difference.

**Table 1 nutrients-15-04161-t001:** Characteristics and quality assessments of eligible meta-analyses evaluating the associations between the KD to intervene in all clinical and non-clinical health problems.

Study, Year (Ref)	Study Disease or Population	Outcome Assessments	No. of Studies (Participants, *n*)	Study Design	Intervention/Comparison	Effect Metrics	AMSTAR 2 Rating^2^
Castellana et al. 2020 [56]	Overweight, obese	Body weight, BMI, WC, RR for discontinuation	12 (695)	RCT, cohort	VLCKD vs. control	MD, RR	Low
Choi et al. 2020 [24]	Overweight or obese, non-type 2 diabetes; overweight or obesity, type 2 diabetes	BMI, Body weight, C-reactive protein, Diastolic blood pressure, Fasting glucose, Fasting insulin, HDL, LDL, Serum creatinine, TC, TG, WC	6 (290)	RCT	LCKD/LCD/KD vs. MCCRD/HCD/regular diet/hypocaloric diet/normal diet/LFD/orlistat therapy plus low-fat diet/	SMD	Low
Muscogiuri et al. 2021 [57]	Obesity	BMI, Body weight, FFM FM, Glycemia, HDL, HOMA-IR, HbA1c, LDL TC, TG, WC	13 (674)	RCT, cohort	VLCKD vs. control/LCD/MD/VLCD	MD	Critically low
Cao et al. 2021 [58]	Endurance athletes	Hrmax, RER, RPE, TTE, VO_2_max	10 (171)	RCT, cohort	K-LCHF vs. HCD/HD	SMD	Low
Bueno et al. 2013 [59]	CV risk factor, healthy, type 2 diabetes	Body weight, DBP, HDL, LDL, SBP, TAG	12 (1470)	RCT	VLCKD vs. LFD	MD	Low
Gibson et al. 2015 [9]	-	Desire to eat, fullness/satiety, hunger, prospective consumption	5 (246)	RCT	KVLED/KLCD vs. control/non-KD	WMD	Critically low
Yang et al. 2021 [60]	Cancer	Fasting glucose, fasting insulin, HDL, LDL, TC, TG, RR for ketosis, RR for adverse events	5 (406)	RCT	LCKD vs. ACS/GD	SMD, RR	Critically low
Lee et al. 2021 [61]	Overweight and obese	BMI, body fat mass, body weight, cardiorespiratory fitness (VO2peak), fasting glucose, HDL, LDL, lean body mass, TC, TG, WC	4 (121)	RCT	Combined exercise, LCKD vs. control/exercise, regular diet/exercise, standard diet	SMD	Critically low
Lee et al. 2021 [62]	Athletes	Body mass	2 (39)	RCT	Exercise, KD vs. control/exercise, regular diet/habitual diet/	SMD	Critically low
Whittaker et al. 2022 [63]	Men’s cortisol and testosterone	Cortisol, total testosterone	16 (168)	RCT	KD/MP-KD HP-KD/KD, long-duration exercise/Short-term, MP-KD, long-duration exercise vs. HCD	SMD	High
Zhou et al. 2022 [64]	Type 2 diabetes	BMI, body weight, fasting glucose, fasting insulin, HDL, HOMA-IR, HbA1c, LDL, TC, TG, WC	7 (526)	RCT	LCKD/LCD/KD vs. LCD/MCCRD/Low glycemic index, reduced-calorie diet/American Diabetes Associations’ “Create Your Plate” diet/HCD	MD	Low
Amini et al. 2022 [65]	Overweight; obese cardiovascular risk factor; overweight, hyperlipidemic; healthy people	BMI, body weight, FFM, FM, LBM, PBF	17 (1550)	RCT	LCD/LCKD/High-fat Atkins Diet/KD/VLCD vs. Low-calorie, high-carbohydrate, low-fat diet/low-fat diet/high-carbohydrate, high-fiber diet/high-protein, Zone Diet/low-fat, nonketogenic, low-carbohydrate diet/non-KD/low-fat, reduced-calorie diet/regular diet, exercise/regular diet/very low fat diet/high unsaturated fat diet/control group/low-calorie diet/moderate carbohydrate, calorie-restricted diet/American cancer society diet	WMD	High
Taftian et al. 2022 [66]	Cancer	BMI, body weight	8 (246)	RCT, NRCT	KD vs. control	WMD	Low
Ashtary-Larky et al. 2022 [18]	Healthy people	BFP, BMI, body mass, FFM, FM	11 (212)	RCT	Exercise, KD vs. exercise, regular diet/Western diet/carbohydrate-restricted diet/non-KD group/control/usual diet/mixed diet	WMD	Low

BMI, body mass index; DBP, diastolic blood pressure; FFM, fat-free mass; FM, fat mass; HDL, high density lipoprotein; HOMA-IR, homeostatic model assessment index of insulin resistance; HbA1c, glycated hemoglobin; HRmax, maximal heart rate during exercise; LBM, lean body mass; LDL, low density lipoprotein; RER, respiratory exchange rate; RPE, perceived exertion; SBP, systolic blood pressure; TTE, time to exhaustion; TC, total cholesterol; TG, triglycerides; VAT, visceral adipose tissue; VO2max, maximum oxygen uptake; WC, waist circumference; KD, ketogenic diet; VLCKD, very-low-carbohydrate ketogenic diets; K-LCHF, ketogenic low-carbohydrate, high-fat; MAD, modified Atkins diet; LCD, low-carbohydrate diet; MCCRD, medium carbohydrate, low fat, calorie-restricted diet; HCD, high-carbohydrate diet; LFD, low-fat diet; VLCD, very-low-carbohydrate diets; HD, habitual diet group; KLCD, ketogenic low-carbohydrate diet; VLED, very-low-energy diet; ACS, American Cancer Society diet; GD, general diet; HP, high-protein; MP, moderate-protein; ref, reference; SMD, standardized mean difference; WMD, weighted mean difference; OR, odds ratio; RR, relative risk; RCT, randomized controlled trial.

**Table 2 nutrients-15-04161-t002:** Characteristics and quality assessments of eligible meta-analyses evaluating the associations between ketogenic diet to intervention of all types of epilepsy.

Study, Year (Ref)	Study Disease or Population	Outcome Assessments	No. of Studies (Participants, *n*)	Study Design	Intervention/Comparison	Effect Metrics	AMSTAR 2 Rating^2^
Sourbron et al. 2020 [67]	Pediatric seizures	RR of SFR, ≥50%	5 (374)	RCT	KD/MAD vs. standard therapy/control group	Proportion, RR	Critically low
Rezaei et al. 2019 [5]	Epilepsy	≥50%, ≥90%, SF	75 (3799)	RCT, cohort	KD/MAD vs. control	Proportion	Low
Martin-McGill et al. 2020 [68]	Drug-resistant epilepsy	RR of ≥50%, SF, treatment withdrawal	7 (566)	RCT	KD vs. usual care	RR	High
Wang et al. 2020 [69]	Dravet syndrome	≥50%	6 (157)	Cohort	KD vs. control	Proportion	Critically low
Sharawat et al. 2021 [70]	Lennox Gastaut syndrome	≥50%, SF	3 (55)	Cohort	KD vs. control	Proportion	Critically low
Lyons et al. 2020 [71]	Epilepsy	≥50%, SF	32 (430)	Cohort	KD vs. control	Proportion	Critically low
Liu et al. 2018 [72]	Intractable epilepsy	≥50%, ≥90%, SF	9 (223)	Cohort	KD/MAD/KD, MAD vs. control	Proportion	Critically low
Pizzo et al. 2022 [73]	Pediatric seizure	OR of ≥50%	7 (413)	RCT	KD/MAD vs. standard therapy	OR	Low
Zhang et al. 2022 [74]	CDKL5-related epilepsy	OR of clinical responder rate, OR of definite responder rate	11 (183)	Cohort	KD vs. control	OR	Low

KD, ketogenic diet; MAD, modified Atkins diet; MD, mean difference; OR, odds ratio; RR, relative risk; RCT, randomized controlled trial; ≥50%, proportion ≥50% seizure reduction; ≥90%, proportion ≥90% seizure reduction; SF, seizure freedom.

## Data Availability

All data included in this umbrella review were extracted from publicly available systematic reviews.

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
