# Peer review of "Ketogenic Diet and Multiple Health Outcomes: An Umbrella Review of Meta-Analysis"

_nutrients, 2023, doi:10.3390/nu15194161_

Round 1

Reviewer 1 Report (New Reviewer)

I congratulate the authors for this paper.

I think this review is a huge work, but the language is highly revisable, particularly in the introduction.

line 33 you cannot write a ketogenic diet increases the risk of ketosis, because ketosis state is the aim of the diet.

Line 47 you repeat the sentence, please correct. I suggest to write shorter sentences to be more friendly for the reader

line 49-50 the sentence is not clear, please rephormulate

line 51-52 "induced" is not the correct term, please correct with "produced from". Right after use the term "generate" instead of produce

line 54 in the sentence "to reduce blood glucose concentration, insulin concentration" please use the conjunction

line 55 use increasing instead of increase

line 81 missing reference about case study. 

line 85 Clarify that you are talking about type 1 diabetes. Which other diseases? please specify

line 195 please correct capital letter after dot or use comma

line 204 please correct square brackets of the references

line 228 the sentence "There were two main types of dietary interventions, KD" is virtually unintelligible, it seems to be incomplete

line 229/265 seem to be missing some letters

line 245 I suggest to rotate the figure 3 to be more frendly

line 282 pay attention to italic letter in "had"

line 299 missing reference for bariatric surgery

line 407 adipocytes release FFA, not TG

line 427 you can add a sentence like "This maximize the oxidation of lipids from endogenous sources"

line 457 your sentence is not clear. Like at line 33, KD have the aim to induce ketosis, why do you talk about risk? Which is the reference of this statement?

line 475 please rephrase the conclusions. I suggest to include sentence such as "KD may be a treatment in cancer..." "rigorous guidelines or scientific positions are still missing..."

line 505 please specify that risk of ketoacidosis is small

The text is not always fluent and there are many errors, anyway the meaning of the article is understandable in its keypoint

I suggest the correction from a native speaker  

Author Response

Reviewer 2 Report (New Reviewer)

This is a well-organized and interesting article. Some points should be improved:

- In the Abstract, the authors should report the scientific database that they use to collect their data.

- Currently, the are several variations concerning KD, e.g., typical/classical KD, medium chain triglyceride(MCT) diet, the modified Atkins died (MAD), and low glycemic index treatment (LGIT) diet. The above should be reported in the introduction section.

- In the methods, the authors should also report if they include only studies with the typical KD or they included and other types of KD

- The table 1 is too long. It should be separated into two tables according to the type of outcomes assessments or the specific study populations.

- The resolution of Table 2 is very low and it should apply vertical layout.

- The tables in pages 6 and 7 should include a title/caption.

-  Again, Table to is too long. The authors should separate this table in more tables according to the type of study populations, e.g., Dravet syndrome, infant epilepsi, pediatric epilepsi, refractory childhood epilepsy, refractory adult epilepsi,.

- A small paragraph with the strengths of the study should be added before limitations.

- The concusion section is quite small. The authors should propose what future studies could be done and if there is a literature ga on this topic.

- Moderate editing of English language revisions id required.

This is a well-organized and interesting article. Some points should be improved:

- In the Abstract, the authors should report the scientific database that they use to collect their data.

- Currently, the are several variations concerning KD, e.g., typical/classical KD, medium chain triglyceride(MCT) diet, the modified Atkins died (MAD), and low glycemic index treatment (LGIT) diet. The above should be reported in the introduction section.

- In the methods, the authors should also report if they include only studies with the typical KD or they included and other types of KD

- The table 1 is too long. It should be separated into two tables according to the type of outcomes assessments or the specific study populations.

- The resolution of Table 2 is very low and it should apply vertical layout.

- The tables in pages 6 and 7 should include a title/caption.

-  Again, Table to is too long. The authors should separate this table in more tables according to the type of study populations, e.g., Dravet syndrome, infant epilepsi, pediatric epilepsi, refractory childhood epilepsy, refractory adult epilepsi,.

- A small paragraph with the strengths of the study should be added before limitations.

- The concusion section is quite small. The authors should propose what future studies could be done and if there is a literature ga on this topic.

- Moderate editing of English language revisions id required.

Author Response

Reviewer 3 Report (New Reviewer)

I would like to congratulate the authors for the excellent scientific rigour used in the overall review and highlight the importance they have given to the different studies published and the biases and errors in the publications, as well as the level of evidence when proposing treatments for different diseases without prior knowledge of their applicability and possible errors in the quality and subjective evaluation of studies that could represent a scientific failure.

Round 2

Reviewer 1 Report (New Reviewer)

My compliments to the authors for this revised version of the manuscript

I have only two comments 

line 55 please remove "by"

line 86 please specify better that the risk of diabetic ketoacidosis in T2D is very small, in accordance with the reference provided ("Even for those with type 2 diabetes where insulin sensitivity in the cells is decreased, diabetic ketoacidosis is rare since these catabolic processes are very sensitive to insulin. However, type I diabetics are at risk for diabetic ketoacidosis since the hormone to stop the generation of ketone bodies is present in low amounts or absent entirely.") 

I think quality of English language is substantially improved

Author Response

Reviewer 2 Report (New Reviewer)

The reised version of the manuscript has significantly improvrd and meets the criteria for publication.

Minir english language editing is required

Author Response

This manuscript is a resubmission of an earlier submission. The following is a list of the peer review reports and author responses from that submission.

Round 1

Reviewer 1 Report

1. I completely don't understand why you point ketosis as a side effect " but as you said as a diet that induces ketosis,”"experienced ketoacidosis four times after using a low-carbohydrate diet 29. "

2. "The diet, firstly proposed in the 1920s for the treatment of diabetes and  " - add proper reference

3. What is exact content (%) of fat, proteins and, carbohydrates?

 4. What do you mean "unrestricted intake of high fat" ?

5. I dont understand why you excluded these publications ?
" if the same population was used in several articles, only the most recent, complete, or largest study was included in the umbrella review. If several meta-analyses examined the same population but included different individual outcomes, we included 112 more than one meta-analysis for an outcome to ensure that all existing individual studies 113 were included in the umbrella review"

6.Change the order .

3.2. KD or KD combined with physical activity among healthy individuals

3.1. KD or KD combined with physical activity and obesity or overweight

7.In this part  physical activity and obesity or overweight is discussed but why diabetes ??

3.1. KD or KD combined with physical activity and obesity or overweight 248

"However, in individuals with T2DM,"

8. What about impact on the triglicerids?

9. " to increase the risk of ketosis on cancer" could you discuss it?

10 . "4.4. KD and cancer" not almost discussed.

11. PRISMA - why records are excluded - based on titles and abstracts -1024

With regards reviewer

Reviewer 2 Report

This is a general review on the effect of the ketogenic diet on the outcomes of different diseases and health conditions. It is a well-conducted review without any major methodological drawbacks. The results are based on the evidence found in the different meta-analyses and are of interest to all researchers in this field.

However, I must say that the tables where the results are presented are not friendly. It is difficult to find and compare the results of each meta-analysis with respect to a specific variable (RER, LDL, glucose concentration, etc.). I would suggest authors try to represent the main results using diamond diagrams, or at least improve the presentation of the results by following a simple 'traffic light' visual indicator, green indicates the intervention is effective amber that there is no difference in the investigated comparison and red that the results suggest the intervention is detrimental or less effective than the comparator.

On the other hand, the summary is wrong about the RER. KD is said to increase the RER and it is the opposite as described in the main text.

Reviewer 3 Report

The authors declare to conduct an umbrella review aimed at outlining the strength of evidence on the efficacy of KD interventions. I do have major concerns regarding the contribution of this manuscript to the field. 

Conditions in which KDTs have been used promisingly ( such as migraine and autism) have not been included in the results. Whether excluded because of inclusion or exclusion criteria adopted, the authors should at list mentioned these conditions  in the introduction. If not arbitrarily considered in the literature search, then this is a relevant bias. Moreover, too much different neurological and non-neurological conditions are put on the same level and this can be confusing for the reader. And there is no suggestion that the ketogenic diet has so far been primarily and primarily approved for epilepsy rather than the other mentioned conditions. Additionally not all epileptic conditions need further evidence to demonstrate KD efficacy, for instance GLUT1DS.

The authors often refer to ‘the risk of ketosis’ where ketosis achievement is usually considered a prerequisite for KD effects to take place. Thus, this phrasing is absolutely misleading: authors should better choose the terms intended for those comments.

The literature regarding epilepsy data is not properly commented . Moreover, the authors should make order among the different epileptic conditions cited (different epileptic syndromes are pediatric epilepsies).

In the paragraph ok KD and cancer data should be better exposed and commented in the following section ( for instance which cancer location? It should be cited in the text).

English language should be revised, please see for instance the verb at line 80

Round 2

Reviewer 1 Report

no comments

no comments

Reviewer 2 Report

In my opinion, this paper has greatly improved with the changes that the authors have introduced when presenting the results and I believe that it is suitable for publication.